# Prognostic Value of Tumor Budding for Early Breast Cancer

**DOI:** 10.3390/biomedicines11112906

**Published:** 2023-10-27

**Authors:** Diogo J. Silva, Gonçalo Miranda, Teresina Amaro, Matilde Salgado, Alexandra Mesquita

**Affiliations:** 1Hospital Pedro Hispano, Local Health Unity of Matosinhos, 4464-513 Matosinhos, Portugalmatilde.salgado@ulsm.min-saude.pt (M.S.); alexandra.mesquita@ulsm.min-saude.pt (A.M.); 2Faculty of Medicine, University of Porto, 4200-319 Porto, Portugal

**Keywords:** early breast cancer, tumor budding, biomarker, epithelial–mesenchymal transition

## Abstract

Background: Tumor budding (TB) is a dynamic process associated with the epithelial–mesenchymal transition and a well-established prognostic biomarker for colorectal cancer. As part of the tumor microenvironment, tumor buds demonstrate increased cell motility and invasiveness. Current evidence demonstrates that high levels of TB correlate with disease progression and worst outcomes across different solid tumors. Our work aims to demonstrate the clinical applicability of TB analysis and its utility as a prognostic factor for patients with early breast cancer (EBC). Methods: Retrospective, single-center, observational study, enrolling patients with EBC diagnosed in a Portuguese hospital between 2014 and 2015. TB classification was performed according to the International Tumor Budding Conference 2016 guidelines. Results: A statistically significant relation was found between higher TB score and aggressive clinicopathological features (angiolymphatic/perineural invasion-*p* < 0.001; tumor size-*p* = 0.012; nuclear grading-*p* < 0.001; and Ki-67 index-*p* = 0.011), higher number of relapses (*p* < 0.001), and short disease-free survival (DFS) (*p* < 0.001). Conclusion: We demonstrate that high TB correlates with shorter DFS and aggressive clinicopathological features used in daily practice to decide on the benefit of chemotherapy for EBC. TB represents a needed prognostic biomarker for EBC, comprising a new factor to be considered in the adjuvant decision-making process by identifying patients at a high risk of relapse and with higher benefit on treatment intensification. Clinical trials incorporating TB are needed to validate its prognostic impact.

## 1. Introduction

Biomarkers are measurable biological parameters that distinguish between normal or pathological conditions [1]. Cancer biomarkers comprise tumor cell features detected in tissue biopsies or body fluids that can be used to stratify prognosis and predict the benefit or toxicity of specific therapeutic interventions [2]. 

In the era of precision oncology, biomarker research and implementation in routine clinical care contribute to the enhancement of tailored therapeutic approaches [3]. Predictive and prognostic biomarkers are essential to define and optimize the therapeutic strategy for each patient, aiming for a maximal therapeutic response with minimal toxicity. A prognostic factor can be defined as a feature associated with clinical outcomes in the absence of therapy [4]. The World Health Organization (WHO) classification of tumors focuses on pathological aspects with histological plus molecular categorizations, integrating established and investigational biomarkers [5].

Tumor budding (TB) is defined as the presence of isolated single cancer cells or cell clusters of up to four cancer cells at the invasive tumor front and has emerged as a promising prognostic biomarker predicting disease progression across several tumor types [6,7,8]. The association between tumor budding, disease progression, and worst outcomes in different solid tumors was first described in 1950 [9].

Tumor buds are part of the tumor microenvironment (TME), being most prominent at the invasive front and associated with epithelial–mesenchymal transition (EMT) [10]. EMT is characterized by cytoskeletal rearrangements, cell motility, increased cell-associated proteolytic activity, and invasion [11]. TB is a dynamic process of tumor cell dissociation that shares some of these biological characteristics, with in vitro evidence showing a prominent role of E-cadherin as a regulator of EMT. 

The role of TB as a prognostic cancer biomarker was first validated for colorectal cancer after the standardization of tumor budding assessment by the International Tumor Budding Consensus Conference (ITBCC) grading recommendations in 2016 [12]. TB is currently included in international guidelines for colorectal cancer, influencing therapeutic decisions for patients with pT1 and stage II colon cancer [13,14,15]. For patients with pT1c, intermediate-to-high-grade tumor budding correlates with node involvement, and it is used to guide complementary surgery following endoscopic resection. In stage II patients, high-grade tumor budding is a poor prognostic factor considered for adjuvant systemic therapy guidance [16]. 

Emerging research suggests that TB has prognostic value for other tumor types, including breast, lung, head and neck, esophageal, gastric, and urogenital cancers. However, the lack of a validated disease-specific scoring system does not allow for wider adoption of tumor budding into the classifications of other cancers.

Despite evidence showing that tumor budding could have a prognostic impact on early-stage breast cancer, there is a lack of clinical evidence demonstrating the clinical applicability of tumor budding analysis and its utility as a prognostic factor to guide therapeutic decisions [17]. This study aims to analyze the prognostic relevance of tumor budding determination for patients with early-stage ductal invasive breast carcinoma.

## 2. Materials and Methods

### 2.1. Study Design and Patient Selection

We conducted a retrospective, single-center, observational study including patients diagnosed with early ductal invasive carcinoma of the breast who underwent lumpectomy or mastectomy in a Portuguese tertiary hospital between 2014 and 2015. The inclusion and exclusion criteria are summarized in Table 1. Medical information was obtained with a retrospective review of electronic health records after approval by the local ethical committee. After patient selection, paraffin blocks and hematoxylin–eosin (H&E) slides were retrieved from the pathology archive.

### 2.2. Patient and Tumor Data

A retrospective analysis of electronic health records was used to obtain data on the patient’s gender, age, date of diagnosis, staging according to the eighth TNM classification of AJCC, adjuvant treatment, disease-free survival, overall survival, and pathology analysis (tumor size, grade, stage, angiolymphatic invasion, perineural invasion, axillary lymph node status, immunohistochemical staining for anti-estrogen receptor, anti-progesterone receptor, and Ki-67 index). Cases with a staining prevalence over 1% for estrogen (ER), progesterone (PR), and a complete membrane staining of more than 10% for HER-2 (3+) or positive via fluorescence in situ hybridization (FISH) were considered positives. Tumors exhibiting an IHC score of 1+ or 2+/in situ hybridization (ISH) not amplified were defined as HER2-low, as per the ASCO/CAP 2018 guidelines update and ESMO expert consensus statement [18,19]. The estrogen and progesterone receptor positivity was assessed using five cut-off values (Group 0: 0; Group 1: 1–24%; Group 2: 25–49%; Group 3: 50–74%; and Group 4: 75–100%). All cases classified as invasive ductal breast carcinoma according to the World Health Organization Cancer Classification and graded using the Nottingham grade scoring system were included.

### 2.3. Tumor Budding Assessment

After patient selection, slides from the surgical specimen colored in eosin and hematoxylin were retrieved from the pathology archives. Each slide was paired with a unique randomly assigned code to the corresponding patient data. To grant validity and reproducibility, a pathologist performed tumor budding analysis according to the International Tumor Budding Conference 2016 guideline [12]. A senior pathologist was responsible for monitoring criteria applications and revising budding assessments. 

A tumor bud was defined as a cluster of one to four tumor cells at the invasive tumor front, and tumor budding was assessed in an eosin–hematoxylin-colored slide using a 0.785 mm^2^ area of the invasive tumor front. The slide with greater budding at the invasive front was selected from pathology archives, and the 0.785 mm^2^ area with greater budding was used for assessment. Tumor budding was classified using the ITBCC 2016 groups (Table 2): low (Bd1), intermediate (Bd2), and high (Bd3).

### 2.4. Statistical Analysis

Analysis for descriptive and inferential statistics was performed using SPSS Statistics^®^ V.23.0 from IBM^®^. Descriptive statistics for each variable were given as frequencies plus percentages of the total. For continuous numerical variables, normality was assessed using Shapiro–Wilk tests plus histogram graphics and reported as a mean ± standard deviation. Variables not following a normal distribution were reported as median (minimum–maximum). Inferential statistics were used to compare numerical variables between the three groups of tumor budding classification according to normality assessment. For those with a normal distribution, a Student’s *t*-test was used, and for those not normally distributed, a Mann–Whitney U test was performed. The relationship between categorical variables was evaluated with Pearson’s chi-squared and Fisher’s exact test, considering the number of patients in the categories. A receiver-operating curve (ROC) analysis was performed to determine the optimal cut-off for the tumor budding score in predicting disease-free survival. Effects of variables on survival were evaluated using the Kaplan–Meier survival analysis and the log-rank test. For statistical significance, the *p*-value was defined as *p* < 0.05.

## 3. Results

### 3.1. Patient Demographics

A total of 100 (100%) patients met the study inclusion criteria, with 98 (98%) female and 2 (2%) male patients. The median age at diagnosis was 63 (33–98) years old, with a median ECOG performance status of 1 (0–3). All patients (*n* = 100; 100%) were diagnosed with no special type (NST) invasive ductal carcinoma of the breast, and the most common clinical stage at diagnosis was Stage I, according to the eighth edition of the AJCC TNM staging system. Classification according to molecular subtypes demonstrated that 35 (35%) patients were Luminal B, 23 (23%) were Luminal A, 19 (19%) were HER-2 low, 15 (15%) were HER-2 positive, and 8 (8%) were triple-negative breast cancers (TNBC). Table 3 summarizes the clinicopathological characteristics of the study population.

All patients underwent surgery, with 69 (69%) undergoing lumpectomy and 31 (31%) undergoing radical modified mastectomy. Following multidisciplinary discussion on systemic treatment, 51 (51%) patients were proposed for adjuvant endocrine therapy, 26 (26%) patients for adjuvant chemotherapy followed by adjuvant endocrine therapy, 9 (9%) patients were proposed for chemotherapy plus anti-HER-2 blockage followed by adjuvant endocrine therapy plus anti-HER-2 blockage, 5 (5%) patients for adjuvant chemotherapy, 4 (4%) patients for adjuvant endocrine therapy (ET) plus anti-HER-2 blockage, 4 (4%) for clinical surveillance, and 1 (1%) for anti-HER-2 blockage.

The median follow-up time was 101 (8–112) months, with disease-free survival of 98.5 (8–112) months. During follow-up, 9 (9%) patients presented disease progression with metastatic disease.

Pathological staging, according to the eighth edition of the AJCC TNM staging system, showed that 64 (64%) patients had pT1 tumors and 36 (36%) pT2 tumors. Regarding nodal status, 65 (65%) patients were classified as pN0, 21 (21%) as pN1, 13 (13%) as pN2, and 1 (1%) as pN3. The histological nuclear grade was classified as Grade 1 (G1) in 42 (42%) patients, Grade 2 (G2) in 35 (35%) patients, and Grade 3 (G3) in 28 (28%) patients. Angiolymphatic and perineural invasions were seen in 30 (30%) patients. For estrogen receptors, 14 (14%) patients were without an estrogen receptor expression, 2 (2%) patients were classified as Group 1, 9 (9%) patients as Group 2, 7 (7%) patients as Group 3, and 68 (68%) as Group 4. For progesterone receptor expression, 40 (45%) patients were without expression, 9 (9%) patients were classified as Group 1, 6 (6%) as Group 2, 14 (14%) as Group 3, and 26 (26%) as Group 4 (Table 4).

### 3.2. Tumor Budding Assessment

The mean tumor budding was 2.36 (±3.69), with 69 (69%) patients classified as low TB, 17 (17%) as intermediate TB, and 14 (14%) as high TB (Table 4).

The relation between tumor budding groups and clinicopathological features is shown in Table 5. A higher number of tumor buds was associated with the occurrence of lymphovascular and perineural invasion (*p* < 0.001), tumor size (*p* = 0.012), higher nuclear grading (*p* < 0.001), molecular subtype (*p* = 0.019), Ki-67 index (*p* = 0.011), and adjuvant chemotherapy (*p* = 0.014). 

There was no statistically significant difference between the pre-defined TB groups for nodal status, estrogen, or progesterone receptor expression.

### 3.3. Tumour Budding and Survival Analysis

Survival analysis showed a statistically significant difference between TB groups for the number of relapses (*p* < 0.001) and disease-free survival (*p* < 0.001) (Table 6). A ROC analysis of TB and disease-free survival demonstrated that a TB score of 5.50/0.785 mm^2^ was the cut-off point for relapse in our population (sensitivity: 0.889; specificity: 0.209; and AUC: 0.839) (Figure 1). No statistical significance was found for the relation between TB and overall survival (*p* > 0.05). 

## 4. Discussion

Pathological changes associated with tumor budding represent the metastatic process in the initial stage, where cells acquire metastatic potential by detaching from the main tumor [20]. The metastatic process relies on features collectively known as epithelial–mesenchymal plasticity that include two major processes: epithelial-to-mesenchymal transition and mesenchymal-to-epithelial transition [21]. In the initial steps of the metastatic process, cells gain mesenchymal features that allow for cell motility plus invasion, and as tumor cells adhere to the metastatic site, they regain epithelial features. The EMT and MET empower the plasticity of stem cells, allowing tumor cells to alternate between epithelial and mesenchymal states and facilitating metastatic spread [22]. 

Tumor buds are part of the TME and are associated with EMT, providing a known portrayal of the initial steps of the metastatic cascade [23]. Histological and molecular subtypings of breast cancer have shown that EMT impacts prognosis, with basal-like breast cancers demonstrating a higher metastatic spread based on sustained mesenchymal features [24]. Transcription factors, including Snail1/Snail, Snail2/Slug, Twist, and the ZEB family of transcription factors, are involved in the EMT process. The upregulation of ZEB is linked with invasive ductal breast cancer de-differentiation, increased vimentin plus N-cadherin expression, and downregulation of E-cadherin [25]. Having a tumor suppressor role in breast cancer, partial or total loss of E-cadherin expression correlates with invasiveness, increased tumor grade, and poor prognosis [26]. 

E-cadherin expression differs between breast cancer subtypes, with lobular subtypes showing a high loss of E-cadherin expression and invasive ductal carcinomas having a low frequency of E-cadherin expression loss [27]. The level of E-cadherin expression influences the invasion pattern, with lobular carcinomas having a single-cell infiltration pattern and invasive ductal carcinomas presenting with solid layers or ductal structures. These patterns justify the reason for the reported tumor budding assessments in the literature being performed exclusively on invasive ductal carcinomas. The solid layer or ductal invasion pattern allows for an easy morphological assessment of TB in H&E slides without the need for additional immunohistochemical characterization [28,29]. 

Evidence implies the role of TB as a prognostic factor for several solid cancers [23], being validated as an independent risk factor for colorectal cancer after the standardization provided by the ITBCC recommendations [13]. Despite some evidence pointing to a similar prognostic role of TB in breast cancer, no validated assessment method exists, and no cut-off values are defined to guide clinical practice. Our study explored the association between TB and clinicopathologic features of early breast cancer, aiming to assess TB’s clinical value as a prognostic biomarker.

Lymphovascular (LVI) and perineural invasion, defined as the presence of tumor emboli within an endothelium-lined space, has been associated with worst outcomes and increased probability of distant metastases in breast cancer [30]. Tumor budding has been established as a predictor of lymphovascular invasion for colorectal cancer [31]. However, the association between TB and LVI for breast cancer is not entirely established, with some research showing that higher scores of budding correlate to a higher extent of LVI [22,29]. Analysis of the surgical specimens presenting with LVI in our population demonstrated a statistically significant association between the TB score and the occurrence of LVI. Of the 30 patients with LVI, 24 (80%) had a TB score greater than 5 buds, and 9 (30%) had a TB score greater or equal to 10 buds (*p* < 0.001). These findings support the current literature on the association between TB score and LVI for breast cancer, with higher scores meaning higher LVI extension and more aggressive biological behavior. 

The histological grade (HG), representing the morphological degree of tumor differentiation, is one of the best-established prognostic factors in breast cancer [32]. Evidence shows that HG can predict tumor behavior, particularly in early small tumors, with high-grade breast cancers tending to recur and metastasize early following diagnosis [33]. Despite representing two different biological processes, evidence points to an association between the histological grade and tumor budding, with higher budding scores associated with higher histological grades and tumor de-differentiation in colorectal cancer [34]. The relation between the histological grade and TB for invasive ductal breast carcinoma is controversial, with some studies showing a trend for higher de-differentiation with higher TB but without statistical significance [35]. Analyzing the distribution of the histological grade via TB group in our study, we found that 93% (*n* = 39) of low-grade tumors (grade 1) presented with a low TB score (0–4) and 75% (*n* = 18) of high-grade tumors presented with an intermediate-to-high TB score (>5 buds) (*p* < 0.001). These findings demonstrate a significant association between the histological grade and TB score, supporting the current evidence for colorectal cancer. 

Tumor size is part of the classical prognostic factors for breast cancer and one of the most important factors in the American Joint Committee on Cancer’s staging system [36]. The association between tumor size and degree of tumor budding has been explored for different solid tumors, with the literature reporting a transversal association between both features [23,37,38,39]. The assessment of mean tumor size across our TB groups showed a significant difference between low and high TB scores, with an increase in tumor size for higher TB scores (low-TB median size: 15.00 (3–49) mm; high-TB median size: 22.50 (8–45) mm; and *p* = 0.012). 

Immunohistochemical evaluation of the Ki-67 proliferation index is critical for the St. Gallen Consensus on differentiating Luminal A and Luminal B molecular subtypes [40]. Being strongly associated with tumor proliferation and aggressive tumor biology, Ki-67 is accepted as a prognostic biomarker for breast cancer and plays a role in therapeutic decisions for breast cancer [41]. Our results are in line with the current literature, showing a significant relation between the Ki-67 index and the degree of tumor budding with an increase in the median Ki-67 index from low to high TB scoring groups (low-TB median Ki-67 index (%): 33.14 (3–90); high-TB median Ki-67 index (%): 51.57 (15–90); and *p* = 0.011).

Previously mentioned features are the basis for defining breast cancer molecular subtype classification, which has an established prognostic and therapeutic impact [41]. Xiang et al. performed a molecular analysis on 240 tumor tissue microarrays and assessed TB scores for each molecular subtype according to the ITBCC recommendations. Results showed that HER-2-positive and triple-negative breast cancers were associated with higher levels of TB and worse outcomes [17,42]. To provide insight into the relevance of molecular subtyping for TB expression, we grouped patients according to their molecular profile in Luminal A (*n* = 23), Luminal B (*n* = 35), HER-2 low (*n* = 19), HER-2 positive (*n* = 15), and TNBC (*n* = 8). Tumor budding analysis stated a significant difference in molecular distribution across TB groups, with a lower number of Luminal tumors (A and B) in the high-TB group (*n* = 6; 10%) compared to HER-2 positive or TNBC tumors (*n* = 6; 25%) (*p* = 0.019). The asymmetric distribution with a higher percentage of HER-2 positive/TNBC in the high-TB group seems to reflect the intrinsic aggressive biological behavior associated with these subtypes of breast cancer.

Regarding prognostic biomarkers for TNBC, tumor-infiltrating lymphocytes (TILs) have a level 1B of evidence to predict clinical outcomes in early TNBC. TILs represent a surrogate biomarker of lymphocyte-mediated immunity, with higher TIL scores associated with higher responses and improved survival outcomes [29,43]. TILs can be characterized according to their distribution in the tumor microenvironment into stroma TILs (sTILs) and intra-epithelial compartment TILs. Recently, the spatial heterogeneity of the tumor microenvironment has been investigated, as subpopulations of tumor cells are unevenly spatially distributed and responsible for different immune microenvironments. Current evidence demonstrates a negative relationship between TB and TILs, with low levels of TB being a surrogate for higher levels of TILs [44]. Therefore, high levels of TILs seem to represent low TB scores and convey a better prognosis. 

Our results point to a relation between higher TB scores and clinicopathological features known to be associated with aggressive biological behavior. This is also supported by the relation found between higher levels of the TB score and the indication for systemic adjuvant chemotherapy. Analyzing the distribution of patients that underwent adjuvant systemic chemotherapy, there was a significant difference between the high TB (*n* = 9, 64%) and intermediate TB (*n* = 11; 65%) groups compared to the low TB (*n* = 23; 33%) (*p* = 0.014). 

These findings illustrate that the subset of high-TB patients has clinicopathological features of greater biological aggressivity that led multidisciplinary tumor boards to propose adjuvant chemotherapy to reduce the relapse rate. We can extrapolate that if no adjuvant strategies were implemented, the difference in disease-free survival of 7 months observed between the low-TB and high-TB groups would be even higher. 

Regarding the impact of tumor budding on survival, the most structured evidence is provided by a post hoc analysis of the IDEA-France phase III (PRODIGE-GERCOR) clinical trial. In post hoc analysis, intermediate and high tumor budding scores strongly correlate with poor disease-free (HR: 1.41, 95% CI: 1.12–1.77; *p* = 0.003) and overall survival (HR 1.65, 95% CI:1.22–2.22; *p* = 0.001) for colorectal cancer. The 3-year DFS and the 5-year OS rates for low TB versus intermediate-to-high TB were 79.4% versus 67.2% (*p* = 0.001) and 89.2% versus 80.8% (*p* = 0.001) [45]. The impact of tumor budding on breast cancer-specific mortality and relapse is controversial, with studies suggesting that there is a decrease in overall and disease-free survival and others not finding any relevant correlation [38,42,46]. Our results showed a significant difference in the number of relapses between the low TB and the intermediate-to-high TB groups, with only one (1.5%) relapse in the first group and eight (26%) relapses in the combined groups (*p* < 0.001). A significant difference was also found for disease-free survival (DFS), with a shorter time for the intermediate (96.00 (8–111) months) and high TB groups (94.00 (21–111) months) versus the low TB group (101 (10–112) months) (*p* = 0.00). No statistically significant differences were seen for overall survival between groups, but data remain immature regarding the follow-up time. 

These findings are in line with the current literature, suggesting a possible impact of TB on disease-free survival for breast cancer patients. 

Despite the results described, the present study has limitations that must be considered when extrapolating our findings to a broader population of breast cancer patients. The study was designed as a retrospective, single-center study enrolling a small number of patients. Furthermore, the tumor budding assessment was performed by two pathologists using ITBCC 2016 guidance, which is not validated for breast cancer, and different results could be found if a different methodology was applied. Finally, despite a median follow-up time of 101 (8–112) months or 8 (0.7–9.3) years, few relapses were documented, which could have an impact on survival analysis.

## 5. Conclusions

Tumor budding is a surrogate for the epithelial–mesenchymal transition process, which is the first step in the metastatic cascade. Its validation as an independent prognostic factor for colorectal cancer following the ITBBC 2016 recommendations led to a growing interest in the prognostic role for other solid cancers. Several studies explored the correlation of different assessment methodologies of TB with clinicopathological features known to influence breast cancer’s natural history and management. Our findings confirm what is currently described, pointing out a statistically significant relation between higher tumor budding scores and clinicopathological features such as lymphovascular/perineural invasion, tumor size, and higher nuclear grading, which are associated with aggressive biological behavior and increased relapse. Furthermore, our study highlights a relation between higher budding scores and referral for adjuvant systemic therapy. These findings show that higher budding scores correlate with a biologically aggressive phenotype corroborated via other clinicopathological features that motivated referral for adjuvant systemic chemotherapy. Finally, the impact of intermediate-to-high budding scores on a higher number of relapses and shorter disease-free survival consolidates its potential as a prognostic factor for early breast cancer. These findings raise the question of whether high tumor budding cases benefit from more intensive treatment regimens with closer surveillance protocols and state the need for using TB as a stratification factor within clinical trials.

Therefore, our study adds to the current evidence that tumor budding is a prognostic biomarker for breast cancer, with further research needed to validate an assessment methodology and explore therapeutic strategies targeting the epithelial–mesenchymal transition process. 

## Figures and Tables

**Figure 1 biomedicines-11-02906-f001:**
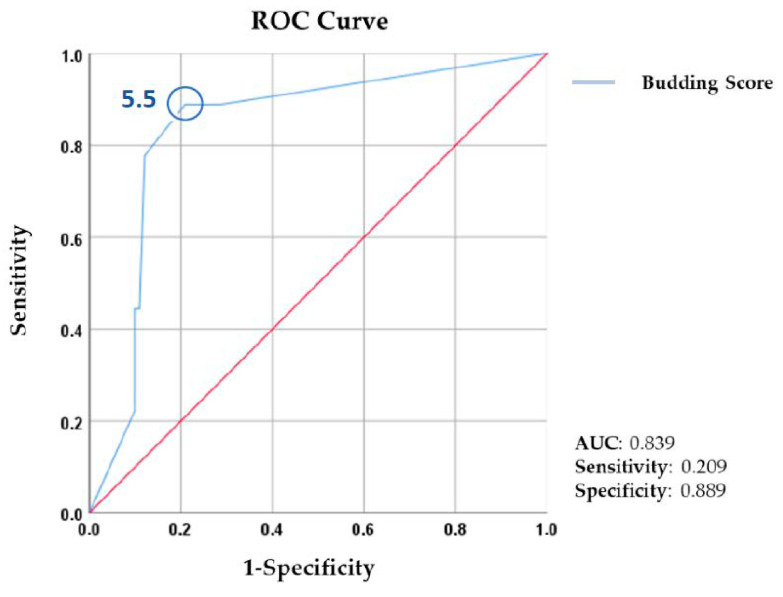
ROC analysis demonstrated that a TB score of ≥5.5 buds was associated with a higher risk of relapse (AUC: 0.839; sensitivity:0.209; specificity:0.889). The red line represents the diagonal reference line. AUC: area under the curve. ROC: receiver-operating characteristics.

**Table 1 biomedicines-11-02906-t001:** Inclusion and exclusion criteria.

Inclusion Criteria	Exclusion Criteria
1. Age over 18 years old;	1. Age less than 18 years old;
2. Histopathological confirmed ductal invasive breast carcinoma;	2. Histopathological confirmed in situ ductal carcinoma or lobular carcinoma;
3. Underwent lumpectomy or mastectomy between 2014 and 2015;	3. Metastatic disease at diagnosis;
4. Early breast cancer at diagnosis.	4. Neoadjuvant systemic treatment for breast cancer.

**Table 2 biomedicines-11-02906-t002:** Definition of tumor budding score groups based on the ITBCC 2016 guideline.

Number of Buds	ITBCC 2016 Classification
0–4 buds	Low (Bd1)
5–9 buds	Intermediate (Bd2)
>10 buds	High (Bd3)

**Table 3 biomedicines-11-02906-t003:** Clinicopathological characterization of the study sample.

Clinicopathological Characterization	
Sample, *n* (%)	100 (100%)
Gender, *n* (%)	
Female	98 (98%)
Male	2 (2%)
Age (median, years)	63 (33–98)
ECOG Performance Status	1 (0–3)
Histological Classification	
NST invasive ductal breast carcinoma	100 (100%)
Surgical Procedure	
Lumpectomy	69 (69%)
Modified Radical Mastectomy	31 (31%)
Staging at Diagnosis	
Tumor Size	
pT1	64 (64%)
pT2	36 (36%)
Nodal Status	
pN0	65 (65%)
pN1	21 (21%)
pN2	13 (13%)
pN3	1 (1%)
Histopathological Characteristics	
Histological Nuclear Grade	
Grade 1	42 (42%)
Grade 2	35 (35%)
Grade 3	23 (23%)
Angiolymphatic and Perineural Invasion	
Yes	30 (30%)
Hormone Receptor Status	
Estrogen Receptor	
Negative	14 (14%)
Positive	86 (86%)
Progesterone Receptor	
Negative	45 (45%)
Positive	55 (55%)
Molecular Classification	
Luminal A	23 (23%)
Luminal B	35 (35%)
HER-2 Low	19 (19%)
HER-2 Positive	15 (15%)
TNBC	8 (8%)
Adjuvant Systemic Treatment	
Endocrine Therapy	51 (51%)
Chemotherapy	5 (5%)
Chemotherapy plus ET	26 (26%)
Chemotherapy plus Anti-HER-2 Blockage plus ET	9 (9%)
ET plus Anti-HER-2 Blockage	4 (4%)
Anti-HER-2 blockage	1 (1%)

**Table 4 biomedicines-11-02906-t004:** Tumor budding assessment.

Tumor Budding Classification	*n* (%)
Low TB (0–4)	69 (69%)
Intermediate TB (5–9)	17 (17%)
High TB (>10)	14 (14%)

**Table 5 biomedicines-11-02906-t005:** Relation of clinicopathological features with tumor budding groups.

	Low TB (0–4)	Intermediate TB (5–9)	High TB (>10)	
Angiolymphatic/Perineural Invasion	6	15	9	*p* < 0.001
Tumor Size (mm)	15.00 (3–49)	19.00 (12–43)	22.50 (8–45)	*p* = 0.012
Histological Grade				
Grade 1	39	1	2	*p* < 0.001
Grade 2	25	6	4	
Grade 3	5	10	8	
Molecular Subtype				
Luminal A	18	3	2	*p* = 0.019
Luminal B	24	7	4	
HER-2 low	13	4	2	
HER-2 positive	11	3	1	
TNBC	3	0	5	
Ki-67 Index (%)	33.14 (3–90)	45.59 (15–95)	51.57 (15–90)	*p* = 0.011
Adjuvant Chemotherapy				
Yes	23	11	9	*p* = 0.014
No	46	6	5	

**Table 6 biomedicines-11-02906-t006:** Survival analysis across tumor budding groups.

	Low TB (0–4)	Intermediate TB (5–9)	High TB (>10)	
Disease Relapse				
Yes	1	6	2	*p* < 0.001
No	68	11	12	
Disease-Free Survival (months)	101 (10–112)	96.00 (8–111)	94.00 (21–111)	*p* < 0.001

## Data Availability

The data presented in this study are available upon request from the corresponding author.

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
