# Peer review of "Prognostic Value of Tumor Budding for Early Breast Cancer"

_biomedicines, 2023, doi:10.3390/biomedicines11112906_

Round 1

Reviewer 1 Report

Comments and Suggestions for Authors

The manuscript titled” Prognostic Value of Tumor Budding for Early Breast Cancer” by Mesquita et al. explored the prognostic role of tumor budding (TB) in early-stage ductal invasive breast cancer.

Major concerns:

 Although the role of tumor budding in breast cancer is novel as pathological changes associated with tumor budding represent the initial stage of metastasis. Cancer metastasis being the key cause of failure of cancer therapy and mortality, finding the biomarkers at early-stage cancer is significant. The study being retrospective with small number of patients and only early-stage ductal invasive breast carcinoma needs further evaluation considering the heterogeneity of breast cancer.  In triple negative breast cancer (TNBC) which is the most aggressive form of breast cancer, lower levels of TB have been found.  Although many key EMT proteins upregulated in TNBC, no correlation between TNBC and high-grade budding was found. 

Minor Changes:

Figure 1 needs to be revised as it is font size and legends are not clear.

Author Response

Dear Reviewer 1, 
We would like to thank you for the relevant comments and suggestions made. Please see the attachment detailing our responses. 

Thank you very much for your time and help.

Reviewer 2 Report

Comments and Suggestions for Authors

This is a well-written manuscript. However, while the information may prove to be important, it is a very small study.  From the references cited, it is clear several other studies examining tumor budding in colorectal cancer also use small sample sizes, but this calls into question the significance of the results. Especially since the results from this study show a minimal drop in the median disease-free months associated with increased in tumor budding (Low budding – 101 months, Intermediate – 96 months, and high – 94 months). Perhaps for overall survival, high tumor budding could be associated with a shorter survival if the low and intermediate budding categories were to be combined. Additionally, tumor budding does not predict disease relapse. While relapse occurred in 1.4% of patient with low budding, 35.3% experienced relapse in the intermediate group and only 14.3% experienced relapse in the high group.  However, the authors are clear that this is a small study, and more analysis needs to be completed to determine whether tumor budding can be used as a biomarker in EBC. While I believe this study to be low impact containing analysis of a small sample set, it could potentially add to future assessments of whether tumor budding determination in EBC has prognostic value.

Power analysis performed by a biostatistician for results confidence could add strength to the study.

It is unclear why information about the adjuvant therapies is included. Is it simply because that information is known? Adjuvant therapy likely does not affect tumor budding since it is given after the surgery. However, neoadjuvant may have the potential to affect tumor budding. Perhaps adjuvant therapy has the most influence on survival, especially if targeted therapy vs chemotherapy was used? Not clear whether it is possible to achieve significant results from stratifying tumor budding with the different chemotherapies in a small sample size. Comments about the adjuvant therapy and its potential, in totality or individually, to affect survival outcomes should be addressed, instead of or at least in addition to, the information about EMT which seems a bit tangential.

While the methods briefly mention how the tumor budding is assessed, it would be helpful to add a reference. In fact, Zlobec et al., 2020 is already in the citations and is likely a good reference choice here as well.

Line 169, I believe table 6 should be referenced, not table 4. Also, there is a random “o” in the line.

Author Response

Dear Reviewer 2, 
We would like to thank you for the relevant comments and suggestions made. Please see the attachment detailing our responses. 

Thank you very much for your time and help.

Round 2

Reviewer 1 Report

Comments and Suggestions for Authors

The authors have revised the manuscript. Figure needs to be revised.  Please use legends to explain the key message of your figure. Use automatic black (instead of grey) for the X-and Y axis to bring reader’s attention. Explain the red bar and bold the abbreviations and symbols.

Author Response

Dear Reviewer 1, 
We would like to thank you for the comprensive review provided. 

Please find attached the point-by-point response. 

Best regards, 
Diogo
